# Chitosan-Based Biomaterials: Insights into Chemistry, Properties, Devices, and Their Biomedical Applications

**DOI:** 10.3390/md21030147

**Published:** 2023-02-24

**Authors:** Simona Petroni, Irene Tagliaro, Carlo Antonini, Massimiliano D’Arienzo, Sara Fernanda Orsini, João F. Mano, Virginia Brancato, João Borges, Laura Cipolla

**Affiliations:** 1Department of Biotechnology and Biosciences, University of Milano-Bicocca, 20126 Milano, Italy; 2Department of Materials Science, University of Milano-Bicocca, 20125 Milano, Italy; 3CICECO–Aveiro Institute of Materials, Department of Chemistry, University of Aveiro, 3810-193 Aveiro, Portugal

**Keywords:** chitosan, marine-origin polysaccharides, chitosan-based materials, hydrogels, layer-by-layer devices, 3D printing, organic–inorganic hybrids, drug delivery, tissue engineering, regenerative medicine

## Abstract

Chitosan is a marine-origin polysaccharide obtained from the deacetylation of chitin, the main component of crustaceans’ exoskeleton, and the second most abundant in nature. Although this biopolymer has received limited attention for several decades right after its discovery, since the new millennium chitosan has emerged owing to its physicochemical, structural and biological properties, multifunctionalities and applications in several sectors. This review aims at providing an overview of chitosan properties, chemical functionalization, and the innovative biomaterials obtained thereof. Firstly, the chemical functionalization of chitosan backbone in the amino and hydroxyl groups will be addressed. Then, the review will focus on the bottom-up strategies to process a wide array of chitosan-based biomaterials. In particular, the preparation of chitosan-based hydrogels, organic–inorganic hybrids, layer-by-layer assemblies, (bio)inks and their use in the biomedical field will be covered aiming to elucidate and inspire the community to keep on exploring the unique features and properties imparted by chitosan to develop advanced biomedical devices. Given the wide body of literature that has appeared in past years, this review is far from being exhaustive. Selected works in the last 10 years will be considered.

## 1. Introduction

Glucose and its 2-acetamido-2-deoxy derivatives are the most abundant organic compounds found on earth, in the form of their β-1,4-homopolymers cellulose and chitin, respectively (Figure 1). Notably, glucose and 2-acetamido-2-deoxy-glucose (namely *N*-acetylglucosamine, Glc*N*Ac) are biosynthetically connected [1], being glucose 6-phosphate their common precursor; similarly, the resulting homopolysaccharides share a role in supporting the integrity, protection and structure of plants (cellulose), or arthropods and fungi (chitin) [2,3].

After cellulose, chitin is the second most abundant polysaccharide on earth. It is estimated that every year 10^10^–10^12^ tons of chitin are produced by living organisms (for more details, the reader may visit the GlycoPedia chitosan section at https://www.glycopedia.eu/e-chapters/from-chitin-to-chitosan/article/abstract-introduction, accessed on 23 December 2022). Over the years, the chitin content has been described in different species, ranging from 75% of dry mass of the exoskeleton in lobsters, to 2% of the dry mass of mycelium in fungi [4], while the degree of acetylation (DA) is commonly 100% [5].

Chitin was first reported as a chemical stable material in 1799 by the English scientist Charles Hachett [6]; however, only a few years later scientists gained interest in the chemistry of this polymer extracted from mushrooms (1811, Henri Braconnot, who named it fungine) [7] and insect exoskeleton (1823, Auguste Odier, who gave it its actual name, chitin). The definite chemical structure of chitin as a polymer made by β-(1 → 4) repeating units of 2-acetamido-2-deoxy-D-glucopyranose was elucidated in 1946 by Earl R. Purchase and Charles E. Braun. CHT was described for the first time in 1859 by Charles Rouget [8], who obtained an acid soluble polymer after boiling chitin in alkali. This soluble polymer was named chitosan in 1894 by Felix Hoppe-Seyler [9]. However, only since the 1970s chitin and CHT begun to be considered as promising materials to be exploited in several applications.

Due to its extensive hydrogen bond network, chitin is totally insoluble in water, as well as in most organic solvents [10,11]. This feature has hampered its characterization [12], and accounts for its chemical and biological stability on the one hand, and its poor industrial application, on the other hand [13].

CHT, which is partially deacetylated chitin in which *N*-acetylglucosamine residues are replaced with glucosamine (2-amino-2-deoxy-D-glucose, Glc*N*), is less widespread in nature and can be found mainly in the cell wall of a few fungi and green algae. Deacetylation converts the homopolymer chitin into a random copolymer of Glc*N*Ac and Glc*N* monomers; the free amino groups can be protonated in acidic media, affecting the physicochemical properties such as solubility, hydrogen bonding and reactivity.

The molecular weight, strictly related to the degree of polymerization (DP), the degree of acetylation (DA) and the pattern of acetylation (PA) are fundamental parameters determining the CHT properties, firstly, in terms of solubility; and secondly, in terms of physicochemical properties and structure-activity relationship [14,15,16].

The average molecular weight of CHT affects its solution viscosity, mechanical, chemical, and optical properties, as well as chain orientation and entanglement. While chitin is reported to have an average molecular weight up to millions Da (in the range 1 × 10^6^–2.5 × 10^6^ Da, that is a DP comprised between 5 and 10 thousands of monomeric units), CHT average molecular weight is commonly comprise between 50 thousand and 1 million Da [17], while most widespread commercial CHT is within the range of a few thousand Da (3800 Da) to 500 kDa [18].

DA, defined as the average number of Glc*N*Ac units per 100 monomers given as a percentage, can be determined with different techniques, including spectroscopic methods [18,19], or potentiometric titrations [20]. When the DA is less than 50%, CHT becomes soluble in slightly acidic aqueous media (pH < 6), and ^1^H-NMR may be the technique of choice for DA determination [21]. Although there is no consensus on the DA value determining the name switch from chitin to CHT, commercial CHT is commonly featured with a DA in the range 2–45% [18]. However, more than a decade ago, the European Chitin Society (https://euchis.org/, accessed on 21 February 2023) proposed the use of the terms chitin/CHT based on the solubility in 0.1 M acetic acid, being chitin is the insoluble polymer and CHT the soluble one [22].

Beside DP and DA, the PA is an additional key feature in assessing CHT properties. Recent studies showcase the relationship between a specific sequence of Glc*N* and Glc*N*Ac monomers and CHT biological activity: a “CHT code” is coming to light [23]. However, while DP and DA have been extensively studied over the last decades, and several techniques developed, PA has been understated mainly because of the lack of methods to obtain defined PA in CHT samples, accompanied by the limited development of specific analytical techniques with suitable sensitivity and robustness [24].

Commercial interest in CHT is due to its numerous features, including biodegradability, biocompatibility, nonimmunogenic, antimicrobial, chelating and polyelectrolyte properties encompassing the food, cosmetic, and pharma industries, and medicine, agriculture and aquaculture, wastewater treatment, textile and pulp sectors.

The industrial chitin and CHT production started in Japan in the early 1970s as a niche material. Currently, society and market demand for renewable feedstocks and new materials in place of fossil-based polymers are boosting industrial interest, and several start-ups and projects are populating the CHT scenario [25].

The CHT industrial market in 2019 was valued globally at 1.7 billion USD, and it is expected to grow at a 14.5% CAGR (compound annual growth rate) from 2020 to 2027, when it is expected to reach a 4.7 billion USD value. The market can be analyzed on the basis of CHT production geographic area, source and application [26]. From a geographical perspective, the Asia-Pacific region is the main production area due to large crustacean consumption, making shell wastes readily available; North America follows, due to the increasing production demand and expanding market of CHT-based biomedical products.

CHT’s main source is seafood industry waste [27], which has been increasing over recent years, and together with the circular economy and waste reuse strategies is supporting the CHT market [28]. CHT is obtained after the deacetylation of chitin found in crustaceans’ exoskeleton, mainly shrimps, crabs, krill, squids (bone plate) and lobsters. Different chemicals and conditions are used for CHT production, depending on the chitin source [29]. Besides the chemical methodologies, a few steps processing chitin into CHT (i.e., deacetylation) can be performed by following more sustainable approaches using biocatalysis [30] or fermentation processes [31,32]. However, biotechnological approaches are still far from being optimized for largescale production [33]. In the future, it is expected that CHT will be obtained from fungi, either as agriculture waste or dedicated fungi cultures, and from snails and other terrestrial mollusks. Cultured fungi CHT production is expected to grow for food and beverage applications due to dietary ethno-religious restrictions [34].

Currently, marketed CHT and its derivatives are mainly used in wastewater treatment, due to its effectiveness in the removal of toxic contaminants. The pharmaceutical and biomedical sector is the second player in CHT industrial applications [35], followed by the cosmetic, and food and beverage sectors, for instance, as a dietary supplement or preservative. Patients and pharmaceutical industries will benefit from the chitosan-based products that will enter into the market, due to their potentialities as drug delivery systems, pharmaceutical formulation components, and active ingredients for different medical treatments. Formulations and drug delivery systems containing chitosan, for example, can help in the delivery of therapeutically relevant active proteins (i.e., growth factors), and peptides, nucleic acids and genes. Patients would benefit in the sectors of regenerative medicine, oncology, dermatology, ophthalmology, dentistry and many others both from a therapeutic and diagnostic point of view.

The use of CHT as a biopesticide and soil supplement for improving plant growth is also expected to grow. More recently, news accounts have highlighted the relevance of circular economy approaches in driving the CHT market. For example, at COP 27, held in Egypt 7–8 November 2022 https://cop27.eg/#/, accessed on 21 February 2023), at the event focused on circular bio-based solutions, Chitosan Egypt (https://chitosaneg.com/, accessed on 21 February 2023) promoted local waste recycling towards chitosan-based bio-pesticides and bio-fertilizers, also entering the market as a CHT producer [32].

The growing interest towards CHT is witnessed by the increasing number of publications and patents since 1990, as denoted in Figure 2. Notably, the number of patents (Figure 2b) exceeds the number of publications (Figure 2a), thus revealing its attractiveness for applied research and industry innovation.

This review aims at providing an overview of CHT properties and chemistry in the design of innovative materials for biomedical applications. Differently from other reviews about chitosan, this review will be organized in sections defined as a function of material typology, rather than as a function of the targeted biomedical application. Strategies towards CHT-based hydrogels, organic–inorganic hybrids, layer-by-layer systems and (bio)inks will be considered organic–inorganic hybrids; layer-by-layer and inks are commonly overlooked in reviews dealing with chitosan. Particular emphasis will be given to the chemistry beyond chitosan material design, highlighting limitations and opportunities. The examples reported here will be far from being exhaustive, given the huge number of published articles; except for seminal papers, the main cited literature spans from 2010 to date.

## 2. Chitosan Functionalities: Friend or Foe?

The switch from chitin to CHT renders the latter less crystalline than chitin and greatly ameliorates its solubility and, hence, reactivity. However, the functional groups present in CHT, which are still able to act as both donor and acceptor of inter/intra molecular hydrogen bonds, are responsible for its water insolubility, unless in slightly acidic conditions (vide infra), thus opening the way to chemical derivatization. The chemical modification of CHT enables tuning its physicochemical and biological properties, ideally embracing any field of application. The chemical functionalization of CHT may involve the two hydroxyl groups, respectively at C-6 (primary) and C-3 (secondary), both on Glc*N* and Glc*N*Ac units, and the 2-amino groups unique to glucosamine units (Figure 3).

Hydroxyl and amino functional groups can react as nucleophiles with electrophilic reagents, being amines more reactive than hydroxyls. Chemoselective and regioselective issues may apply in CHT modification; this lack of selectivity may afford heterogeneous mixtures of derivatives, limiting reproducibility and applications; in light of these observations, robust chemistry allowing the strict control over selectivity is needed for CHT derivatization. Additionally, CHT functionalization is more effective in homogenous conditions, usually requiring aqueous acids as solvents: in these conditions, it should be taken into account that water may be a competing nucleophile in the reaction, lowering derivatization yields. At the same time, acidic pH may promote reagent decomposition. Moreover, at acidic pH, usually needed for chitosan solubilization, amine groups are partially converted into the corresponding ammonium ions, thus reducing their nucleophilicity. On the other hand, CHT derivatization in the heterogeneous phase allows the use of non-nucleophilic solvents and non-acidic pH. However, the degree of substitution is usually low and harsh conditions (i.e., high temperature, excess of reagents) may be required, thus reducing the control over chemo- and regioselectivity, affording structurally heterogeneous mixtures of products, possibly containing degradation byproducts as well [10,36].

The specific chemical nature of the alkyl or acyl group of the derivatizing agents allows the introduction of several functional groups, resulting in the fine tuning of the physicochemical properties of the CHT derivatives [37]. For example, the introduction of hydrophilic groups [38], such as carboxylic [39], sulfates and sulfonates [40], or phosphate groups [41], which at suitable pH are in their anionic form, allows to obtain CHT derivatives, with properties variable as a function of the substitution’s degree. As the main result, these derivatives may become soluble in water at neutral pH; and have better coordination ability towards metal cations, tunable viscosity, hydrogel forming ability, and different biological activity spanning from anticoagulant to antibacterial effects.

When hydrophobic groups are added to CHT, for example by acylation with fatty acids or alkylation with hydrocarbon chains, amphiphilic derivatives can be obtained, useful for their surface active properties and aggregation behavior, affording nanoparticles, micelles and hydrogels [42,43,44,45]. These derivatives find interesting applications in the biomedical field as drug delivery systems and for lipophilic water contaminants absorption (i.e., oils). Additionally, hydrophobic derivatization increases both antibacterial and hemostatic activity [46].

If the derivatization reagents contain sulfhydryl groups (-SH), thiolated CHT derivatives can be obtained [47,48]. Thiol groups may be useful as adhesive groups and antibacterial properties, or their chemical reactivity can be exploited for further chemoselective derivatization reactions through click chemistry approaches [49] or by oxidation to the corresponding disulfide adducts [50]. Disulfide bond formation may be a limitation as well, if undesired.

Taking advantage of the higher nucleophilicity of the amine groups (as the free base) when compared to the hydroxyl groups, several strategies can be applied for the chemoselective chitosan functionalization (Figure 1). The amine group can act as a nucleophile in nucleophilic substitution to (per)alkylated ammonium derivatives (Figure 1a) [51], nucleophilic opening of epoxides (Figure 1b) [52] or aziridines (Figure 1c) [53], nucleophilic acyl substitution to the corresponding amides (Figure 1d) [54], donor in 1,4-Michael additions (Figure 1e) [55], Schiff base adducts formation (Figure 1f) [47,56], eventually followed by reductive amination to the corresponding secondary or tertiary amines (Figure 1g). Notably, the nucleophilic attack of carbonyl derivatives towards the formation of imines (Schiff base) or amines (reductive amination), as well as Michael additions, is fully chemoselective, allowing the derivatization of the sole C-2 position of CHT, while nitrogen peralkylation affords ammonium derivatives, characterized with a pH-independent polycationic character [57].

A robust strategy for fully chemoselective and effective *N*-acylation was proposed by Másson et al., through a Design of Experiment approach (DoE), affording the synthesis of CHT conjugates with cinnamic, *p*-coumaric, ferulic and caffeic acid, with substitution degrees ranging from 3 to 60% [58]. Cinnamyl moieties confer interesting properties when grafted onto different materials, such as antioxidant and antimicrobial properties [59] or exploited for light-triggered processes [60]. The synthetic strategy involves the reaction of 3,6-di-*O*-*tert*-butyldimethylsilyl CHT with *tert*-butyldimethylsilyl-protected acyl chlorides (except for cinnamoyl chloride), followed by an acidic deprotection step. The antimicrobial activity of cinnamoyl CHT resulted analogously in the unsubstituted CHT for the low substituted conjugates, while the higher the substitution, the lower the antibacterial activity. This observation is ascribed to the loss of quaternized primary amino groups. On the other hand, the antioxidant activity is not shown by the cinnamic acid derivatives, but it is strongly promoted by hydroxy cinnamic acid moieties, with the best performance being accomplished with caffeic acid conjugation, affording antioxidant activity 4000 times higher than pristine CHT.

Chemoselective *O*-derivatization over *N*-substitution can usually be achieved after cumbersome protection/deprotection steps. The most commonly used group for selective *N*-protection is the phthaloyl group [61,62], and its derivatives containing electron withdrawing groups [63], which can be deprotected in mild conditions (Figure 2).

Additionally, the *O*-derivatization adds a regioselectivity issue, since both the 6-OH and/or the 3-OH groups may undergo the reaction, affording product mixtures. Depending on the reaction conditions and derivatization agents, 6-OH may be derivatized with fair regioselectivity due to its reduced steric hindrance, when compared to the secondary 3-OH. The favored access to 6-OH can be exploited for its regioselective protection (Figure 3), thus opening the way to selective 3-OH derivatization; however, multistep protection/deprotection steps are needed.

Several reactions can be performed on the hydroxyl groups (Figure 4), including etherification and acylation (Figure 4a,b and 4c, respectively) [64], azidation (Figure 4d) [65], sulfation and sulfonation (Figure 4e) [66], phosphorylation (Figure 4f) [67].

Chemo- and regioselectivity in CHT functionalization may be obtained via biocatalytic processes [68]. Interesting approaches affording non-random *N*-acylation patterns start from polyglucosamine polymers, i.e., fully deacetylated chitin, that can be chemo- and regioselectively amidated to chemically defined CHT through chitin deacetylase [69]. Oxidases [70] are interesting enzymes for the chemoselective amine derivatization with phenolic compounds (Figure 5, inset): gallates, cinnamyl derivatives (i.e., caffeic acid, chlorogenic acid), ferulic acid, hexyloxy phenol, menaquinone, quercetin, and tyrosine containing (macro)molecules have been chemoselectively introduced as *N*-substituents. Different oxidases, such as tyrosinases [71], laccases [72], or horseradish peroxidases [73] were proposed (Figure 5) [74]. Most of these derivatives have antioxidant and antimicrobial properties, thus improving the biological activity of CHT with features that can be exploited in both the biomedical and food packaging sectors.

CHT can also act as a macromer in the synthesis of different copolymers, where the additional macromer component(s) may be of natural or synthetic origin [75]. The chemical nature of the co-macromer often dictates the synthetic strategy [76], and the final physicochemical, biological, and biodegradation properties [77] and applications [78]. Thus, proteins and peptides, poly- and oligosaccharides, including cyclodextrins, and several synthetic macromers (polyacrylates, polyethylene glycol, polyvinyl alcohol, polylactic acid, polycaprolactone etc.) have been grafted to CHT. CHT copolymer synthesis may be accomplished either through “grafting to”, “grafting from” [79], or cross-linking approaches (Figure 6) [80,81]. Regardless of the approach, a key issue in the synthetic design is the complementary reactivity of CHT functional groups (the already in place -OH and -NH_2,_ or those introduced *ad-hoc* by derivatization) and those on the monomer/macromer towards the desired co-polymer. As previously stated, while CHT functionalities offer grafting sites, regio- and chemoselectivity issues limit the synthesis of chemically defined co-polymers. Given the higher reactivity of the C-2 nitrogen, *N*-grafting or *N*-cross-linking usually allows the synthesis of copolymers with well-defined structure and high yields. The use of click chemistry [82,83,84] may further improve the outcome. On the other hand, selective *O*-grafting requires additional protection/deprotection steps of CHT macromers.

In this section, a general outline of the chemistry beyond the CHT functionalization was given, with particular emphasis on regioselectivity and chemoselectivity. The reader may refer to other reviews for an in-depth understanding about specific syntheses of CHT derivatives [17,64,85,86,87,88].

## 3. Modulation of Chitosan Solubility, and Potential Application in the Material Science Field

The type and degree of derivatization, including the degree of acetylation, molecular weight, polydispersity, and the source and chemical treatment for production are key factors modulating the CHT physicochemical properties and applications [30,89,90].

The CHT structure inherits from chitin a semicrystalline structure held together by a strong hydrogen bonding network, which accounts for CHT insolubility in water at neutral pH [91], and in most common organic solvents [14]. In the case of chitin, the hydrogen bonds between the carbonyl oxygen as acceptors, and the amide -NH- groups as donors generate crystalline fibrils that determine the three chitin allomorphs (α, β, γ-chitin) detectable by X-ray crystallography, whose patterns have relevant variation in the chain orientation [92]. Through the process of deacetylation, which weakens hydrogen bonding, CHT partially loses its original crystalline structure and maintains only a semicrystalline form. Differences in solubility were observed between α- and β-CHT, due to the modification of the chain arrangement and its interaction with solvent molecules, affecting the swelling process and stability of the filaments [93]. Besides, differences in the molecular weight and distribution of the acetyl groups along the polymer chain play a role in the CHT crystallinity and, therefore, solubility. These aspects, together with the intrinsic variability of a biologically derived product, make it more difficult to measure the CHT solubility [94]. Residual acetyl groups are deemed to favor polymer chain aggregation and packing owing to their hydrophobicity, while free amine groups (p*K*_a_ = 6.3) [95] are responsible for extended intramolecular and interchain hydrogen bonding, which increases with the polymerization degree, thus accounting for water insolubility. The reduction of these non-covalent interactions either by deacetylation, acid depolymerization, and/or free amine protonation renders CHT soluble in dilute acids and eventually in water. In acidic media, CHT becomes protonated and, thus, positively charged, which turns it into a soluble biopolymer. It has been reported that the solubility is reached when the protonated amines are more than 50% [14]. Since the solubility is closely linked to the degree of charge and, hence, protonated amine groups, the lower the DA, the higher the solubility. Aqueous organic acids are usually used to solubilize CHT, being acetic acid the most commonly used solvent with a concentration ranging between 1 to 5% (pH 4). Other carboxylic acids effective in CHT dissolution are formic and lactic acids. CHT is also soluble in aqueous inorganic acids, such as hydrochloric and nitric acids, despite being insoluble in phosphoric and sulfuric acids.

The insolubility of CHT in water at neutral (physiological) pH is still a limitation in its derivatization and application, especially in the biomedical field. However, new chemical strategies are emerging to enable the dissolution of CHT in a neutral aqueous environment [96]. For instance, to circumvent the lack of solubility at neutral pH, CHT has been modified to include either ionic or highly hydrophilic groups, such as carboxylic, sulfate or *N*-alkyls [97]. The CHT amine functional groups can be derivatized into a quaternary ammonium salt, obtaining a permanent cationic charge, which has shown to improve water solubility, antimicrobial activity and mucoadhesiveness [98]. CHT with an increased solubility can be obtained also via carboxylation by introducing acidic groups into its main chain. Derivatization may increase the CHT solubility up to neutral and alkaline pH, increasing moisturizing and film-forming properties [90].

Since the solubility of CHT is highly desired towards enabling its processing and applications, several approaches have been proposed. For instance, the rupture of intra- and interchain hydrogen bonds triggered by gelation has been proposed as a strategy for enabling the dissolution of CHT, for example, in alkali-urea aqueous solution at low temperature [99]. In this system, the gelation effect is driven by the interaction between amine and hydroxyl groups of the polymer with hydroxide ions in solution, which become very competitive in the formation of hydrogen bonds. The newly established equilibrium between intra/intermolecular hydrogen bonds and hydroxide ions is driven in favor of hydroxide ions in a basic environment, leading to the solubilization of CHT. Urea acts as both hydrogen-bonding acceptor and hydrogen-bonding donor, contributing to the establishment of a new equilibrium where intra/intermolecular interactions among CHT chains are unfavored. Besides aqueous alkali-urea solutions, other alkali aqueous solvents were proposed for the solubilization of CHT, namely LiOH, NaOH, KOH [100].

The dissolution of CHT in organic solvents (i.e., DMF, pyridine, DMSO, dichloromethane, chloroform, or acetone) may be achieved through its derivatization into suitable derivatives [101]; fatty acid side groups were introduced onto the CHT backbone for shifting its wettability from hydrophilic to hydrophobic and even superhydrophobic, towards the development of sustainable coatings [102,103].

Besides solubility, the rheological, surface tension [45,104], adhesiveness [105,106], and self-assembling properties [107,108] can be tuned by CHT derivatization. Hence, the modulation of CHT properties could be achieved by the introduction of new functional groups, thus enabling its use for addressing several applications, including antibacterial [109] and anticancer agents [110], catalysts [111], adsorbents of organic [112] and inorganic pollutants [113], sensors [114], the stationary phase for chromatography [115], surfactants [44,116], and a wide array of biomedical applications [64,117].

All the chemical functionalizations highlighted in Section 2 open up a broad perspective for the development of CHT-based materials and coatings. In a recent paper, Tagliaro et al. [103] fabricated superhydrophobic fluorine-free CHT coatings by the chemical modification of CHT with fatty acid side groups followed by its deposition through a solvent-free method. An optimal balance was also identified to combine hydrophobicity and transparency (Figure 4a,b), showing good durability in abrasion resistance tests, in water and acidic environments and over adhesion tape tests, although further improvement is required to increase the material adhesiveness to the substrate. This study aims at safe- and sustainable-by-design coatings with enhanced functionalities by replacing fluorinated substances, which raise concerns for their potential hazard on human health [118].

CHT film coatings have also shown to increase the shelf-life of fresh products and slow down fruit decay owing to its antifungal and antimicrobial properties. Edible CHT films have been formulated both as pure CHT films or as blends with other polysaccharides, such as starch or alginate, or with proteins, such as milk and soy proteins, collagen or gelatin [120]. Vu et al. [121] have reported the efficacy of a hydrophobized palmitoyl chloride-functionalized CHT film coating in preventing discoloration and decay of fresh products, potentially doubling their shelf-life. Quintana et al. [122] showed that the addition of licorice root extracts to CHT improved the rheological properties of edible coating, as demonstrated by tests with strawberries. It was shown that the fruits not only maintained good quality parameters during storage but also showed the best microbiological preservation in comparison with controls. As such, film edibility enables the direct application of CHT as a coating for food applications (Figure 4c,d) [119], with the potential of reducing and minimizing packaging-related waste.

Besides the potential use of CHT in the food industry, CHT based-materials may also be useful to control the freezing process (nucleation, ice crystal growth and ice aging) of food for below-freezing storage. In general, polysaccharides are known to be surrounded by a hydration shell, where water molecules foster interactions with the polysaccharide functional groups (hydroxyl, -OH, and amine, -NH_2_) [123]. When polysaccharides are cooled down to low temperatures, non-freezable water may be still present [124]. Non-freezable water is a non-crystalline state, which derives from the presence of both free and bound (coordinated) water molecules within the polysaccharide hydration shell (1 to 100 nm). Studies on polysaccharides with low [125] and medium molecular weight (order of 100 kDa) [126] have shown that they can inhibit ice crystal formation and growth due to micro-viscosity, gelation, and hydration. However, more fundamental studies are needed to confirm the potential of CHT to control the freezing process of water and hydrated food.

The possibility to efficiently modify CHT, greatly relying on its solubility in a reaction medium suitable for its chemical modification, renders this biopolymer suitable for multiple applications. In the next sections, insights into the processing of CHT into a wide array of CHT-based materials and devices suitable for biomedical applications are given.

## 4. Chitosan-Based Hydrogels for Biomedical Applications

Hydrogels based on CHT are gaining considerable interest for biomedical applications due to its biocompatibility, biodegradability, non-toxicity, and biological features, such as bio-adhesiveness, antibacterial, hemostatic, and anti-inflammatory properties. Altogether, these properties are pivotal in the design of biomedical systems for smart drug delivery [127,128,129,130,131,132,133,134,135], wound healing [136], and regenerative medicine [137].

Hydrogels are composed by hydrophilic macromers able to capture a large amount of water molecules without dissolving [138], suitable for mimicking the extracellular matrix (ECM) of biological tissues. The capacity to absorb water is achieved through the equilibrium between cohesive and osmotic forces, and depends on the chemical nature of the macromers, cross-linking strategies [76] and 3D structure. Since CHT is a linear polymer, cross-linking is usually needed to obtain hydrogels with suitable mechanical properties. Several synthetic strategies were applied to achieve the required functionalities [139]. Cross-linking can be obtained through physical interactions based on ionic and/or electrostatic forces [140,141] or by covalent bonds [142]. Moreover, regardless of the cross-linking approach, CHT-based hydrogels may be obtained as blends with different natural or synthetic macromers.

Following an appropriate cross-linking chemistry, hydrogels can be designed to respond to an external stimulus, such as light [143], temperature [144], pH [145], and electromagnetic field [146], and are usually referred to as smart or responsive hydrogels [147].

### 4.1. Chitosan-Based Hydrogels by Physical Cross-Linking

Physical cross-linking is based on non-covalent forces, such as ionic bonds, dipole-dipole, ion-dipole, Van der Waals, and hydrophobic and hydrophilic interactions. Physical hydrogels are characterized by weaker mechanical and chemical stability than chemical ones, thus being easily damaged. However, there are applications where the weakness and progressive degradation of the hydrogel is an advantage. This is the case of wound dressing applications, in which it is possible to notice a recurrence of physical hydrogels. The focus on hydrogels for wound dressings is growing since the materials traditionally used, such as gauze and bandage in cotton or wool, have several drawbacks, including low permeation to oxygen, low ability of wound drainage, limited adhesion and being painful and dangerous during removal. Innovative wound dressing materials are designed to address these issues; in this framework, hydrogels offer the advantage of their degradation ability. Additionally, hydrogel properties for damaged tissue regeneration may be improved owing to their ability to act as delivery vehicles of biomolecules for promoting tissue repair (i.e., growth factors, immunomodulators, glycans) [148,149,150], and limit bacterial infections (i.e., acting as anti-bacterial agents). The wound dressings based on CHT hydrogels are intrinsically biodegradable and antimicrobial [151], rendering them suited for wound healing and tissue regeneration.

Given the polycationic nature of CHT at acidic pH, a straightforward physical cross-linking can be achieved by ionic interactions with negatively charged (macro)molecules. Notably, anionic low molecular weight physical cross-linking agents such as phosphate and carboxylate functional groups have been explored for the cross-linking of CHT biopolymer (Figure 5).

CHT-based hydrogels can be obtained with trisodium phosphate (TPS) [152], pyrophosphate (PPi) [153], triple polyphosphate (TPP) [144], or 6-phosphogluconic trisodium salt [154].

In the synthesis of phosphate-based crosslinked hydrogels, the production of a homogeneous 3D matrix represents a major issue. For example, Sacco and co-workers showed that the local quick saturation of binding sites results in the formation of agglomerates, and inhomogeneous chemical and mechanical properties of the hydrogel [155]; however, a controlled diffusion of TPP anions in a CHT solution by means of a semipermeable membrane allows the production of a homogenous hydrogel with good mechanical properties and non-cytotoxicity. Notably, TPP and PPi impart different properties on the hydrogels, the first one promoting the formation of homogeneous materials, and the second affording inhomogeneous hydrogels [156]. Moreover, the morphological analysis of the hydrogels obtained with TPP (Figure 6b) or PPi (Figure 6c) shows that TPP-crosslinked CHT possesses increased connectivity when compared with that of PPi-crosslinked, as revealed by TEM (Figure 6a). The degradability in physiological conditions, diffusion coefficient and drug release behavior reflect the different connectivity and homogeneity features of the systems [153].

6-Phosphogluconic trisodium salt was proposed for the first time as an anionic cross-linker by Martínez-Martínez et al. [154]. This low molecular weight heterobifunctional dianion can be obtained from 6-phosphogluconic acid, an intermediate in the pentose phosphate pathway involving the degradation of glucose. The obtained hydrogel is stable at neutral pH and degrades at pH below 4.5. The swelling properties, and loading and release kinetics can be tuned as a function of the amount of crosslinking agent, affording interesting features for tissue regeneration in wound healing applications.

Thermoresponsive CHT-based hydrogels can be obtained owing to polyols monophosphates [157], such as β-glycerol phosphate [158,159], glucose 1-phosphate [160] or 6-phosphate [157]. The gelation mechanism is not based on cross-linking interactions among amino and phosphate groups as observed for dianions, but instead by the modulation of hydrogen bonding networks among the polysaccharide, polyol moieties and water molecules [157].

An injectable pH-sensitive hydrogel based on carboxymethyl CHT and oligomeric procyanidin was also proposed [161]. Oligomeric procyanidins are natural flavonoids obtained from grape seeds, showing antitumor and antibacterial activity, able to act as a dynamic cross-linker through hydrogen bonding (Figure 7).

The resulting hydrogel showed antibacterial, adhesive, self-healing, and injectable properties, being useful in regenerative medicine applications (Figure 8).

Besides the use of small molecules for physical cross-linking, macromers such as anionic polysaccharides or proteins can also be used. Those include alginate [162], pectin [163], xanthan [164], or hyaluronan [165]. However, the inhomogeneity of the resulting hydrogels is still an issue; advancements towards improving the homogeneous distribution of electrostatic forces have been achieved through acidification of the mixture in the vapor phase, for example by resorting to CHT and pectin [166]. CHT-pectin hydrogels were optimized for 3D-printed scaffold production for regenerative medicine [167].

Proteins such as collagen and its hydrolyzed product gelatin [168] and elastin [169] have been blended with CHT for hydrogel production since they are particularly suited for tissue regeneration, fulfilling two fundamental requisites: favorable interactions with cells (i.e., adhesion) and cytocompatibility [170]. However, proteins do not have a polyanionic character as featured by some polysaccharides, and different non-covalent interactions are brought into play in the hydrogel formation, resulting in limited mechanical properties and stability. Chemical cross-linking is then the most suitable strategy to obtain stable CHT-protein hydrogels.

Physical cross-linked hydrogels may comprise a blend of CHT and other natural macromers. However, synthetic macromers have also been used [140]. Synthetic macromers offer several advantages, including better control over chemical structure and composition, improvement and tailoring of the mechanical properties, and stability against chemical or metabolic degradation. The chemical nature/functionalities of the macromers influences the interactions with biological systems. Cell adhesion is hampered by highly hydrophobic materials with low surface energy, that are unable to establish interactions with cell membrane proteins [171]. Likewise, extremely hydrophilic materials are highly hydrated inhibiting cell adhesion [172].

Synthetic macromers are particularly attractive to address the limitations of low stability and mechanical properties through CHT-based interpenetrating (IPN) and semi-interpenetrating (SIPN) polymer networks. In IPN and SIPN, the hydrogels are still based on non-covalent interactions., However, increased network stability is achieved through macromer interpenetration. Usually, at least a synthetic polymer is needed to achieve the molecular entanglement. IPNs are defined by IUPAC as “polymers comprising two or more networks that are at least partially interlaced on a molecular scale but not covalently bonded to each other and cannot be separated unless chemical bonds are broken (Note: a mixture of two or more preformed polymer networks is not an IPN), while semi-interpenetrating polymer networks (SIPNs) are “polymers comprising one or more polymer networks and one or more linear or branched polymers characterized by the penetration on a molecular scale of at least one of the networks by at least some of the linear or branched macromolecules” (Note: an SIPN is distinguished from an IPN because the constituent linear or branched macromolecules can, in principle, be separated from the constituent polymer network(s) without breaking chemical bonds; it is a polymer blend) [173]. Several synthetic strategies can be used, mainly based on sequential or simultaneous approaches (Figure 7).

For example, CHT-based IPN based on the simultaneous cross-linking of CHT and gelatin by the natural cross-linker genipin were proposed (Figure 8) [174]. Genipin, a compound extracted from the *Gardenia jasminoides*, reacts in a complex multistep sequence with the primary amines of CHT and amino acid sidechains in gelatin, affording different adducts (Figure 8). The resulting hydrogel possesses pH-responsiveness, useful for controlled drug release in biomedical applications.

CHT-based SIPNs have also been proposed, showing improved mechanical and biological properties. For example, CHT was cross-linked with glutaraldehyde in the presence of bacterial cellulose (Figure 9) [175]. The resulting hydrogel showed promising thermal stability, mechanical resistance and flexibility. The modulation of the amount of cellulose with respect to the CHT allowed finetuning of the elastic modulus: the higher the cellulose percentage, the lower the elastic modulus. Additionally, the CHT content modulates the antibacterial properties of the final hydrogel: the higher the chitosan percentage with respect to cellulose, the better the antibacterial activity.

SIPN also have wide application in ex situ cell culture, as in the case of polymethacryloyl glycylglycine (poly-MAGG) cross-linked by radical polymerization with ethylene glycol dimethacrylate (EGDMA) in the presence of CHT [176], being promising for tissue engineering and controlled drug delivery applications.

### 4.2. Chitosan-Based Hydrogels by Chemical Crosslinking

Chemical cross-linking creates stable covalent bonds within macromers, imparting them with higher chemical, biochemical and mechanical stability. However, these features may cause reduced biodegradation and toxicity issues. In fact, the toxicity of the reagents and byproducts is frequently underscored, especially due to the intrinsic inability to wash them out of a gelled matrix [177]. Moreover, as detailed in previous sections, the CHT physicochemical features contribute to amplify the obstacles of chemical cross-linking. Glutaraldehyde [178], genipin [179], and long-chain cross-linkers with suitable reactivity, such as the polyethylene glycol (PEG) diacid [180] have been widely used as agents for enabling the cross-linking of CHT amino groups. In this case, the CHT-PEG cross-linking mechanism is attempted by reacting the glycol diacid with CHT via carbodiimide coupling chemistry. The resulting hydrogel denoted self-healing ability and enabled drug delivery for treating harmful chronic ulcers. The reticulation with a long chain cross-linker is enforced by the dynamic intermolecular interactions due to hydrogen donors and acceptor atoms (nitrogen and oxygen) present in the system.

Considering the risk of byproduct toxicity, emerging cross-linking strategies involve enzymatic reactions [181]. Siqi Zhou et al. [182] reported on the functionalization of CHT with 3,4-dihydroxyhydrocinnamic acid, followed by oxidative cross-linking through horseradish peroxidase/hydrogen peroxide (HRP/H_2_O_2_) (Figure 10).

The obtained hydrogel was studied for its ability to regenerate cartilage tissue. Bone-derived mesenchymal stem cells (BMSC) laden-hydrogel was able to induce chondrogenic differentiation and hyaline cartilage production in vivo, being a promising scaffold for cartilage tissue repair. The repair of articular cartilage tissue is not trivial owing to the lack of a natural regenerative mechanism. The most commonly used approach is usually based on the isolation of autologous chondrocytes or stem cells seeded in 3D scaffolds. Moreover, several studies identified CHT as a suitable biopolymer for cartilage tissue regeneration [183].

CHT has been used for the design of pH-responsive injectable hydrogels loaded with doxorubicin for chemotherapy against colon cancer, and with the antibiotic amoxicillin against *E. coli* and *S. aureus* [184]. In order to obtain a pH-responsive hydrogel, dihydrocaffeic acid moieties have been grafted to CHT through the carbodiimide chemistry to obtain a suitable hydrogel for drug delivery. The CHT-caffeic acid macromers were cross-linked with oxidized pullulan (a polysaccharide constituted by α-1,6-maltotriose repeating units) by Schiff base reaction (Figure 11).

The grafting with dihydrocaffeic acid has multiple purposes: improved adhesion to mucus membranes, higher solubility of CHT and pH responsiveness due to the oxidative cross-linking of the catechol moieties promoted by a pH switch from acidic to physiological media [185]. Pullulan showed to significantly improve the mucoadhesive properties when compared with the pullulan-free CHT-dihydrocaffeic acid system.

A key issue in regenerative medicine is the design of scaffolds promoting and sustaining cell adhesion, growth and differentiation, mimicking as much as possible the native ECM composition, structure, and biochemical and biomechanical stimuli. Donati et al. proposed a mechano-responsive hydrogel based on commercially available Chitlac^®^ [186], obtained from CHT by reductive *N*-alkylation with the reducing end of lactose. Chitlac^®^ was cross-linked with boric acid in different concentrations (Figure 12).

The resulting hydrogels (Figure 9) showed increased viscosity upon thermal or mechanical energy application, due to a rearrangement of the molecular network induced by the boric acid [187]. The dynamic boric ester crosslinks generate a nonequilibrium status where anchoring points are continuously created and destroyed, as a consequence of the adaptation to stress variation [188].

Dynamic chemical bonds, such as imine formation/hydrolysis, accompanied by structural elements favoring hydrophobic intermolecular interactions (i.e., aryl moieties) are the basis for the preparation of thermoresponsive CHT-based hydrogels through a combination of physical and chemical cross-linking. In this regard, natural aldehydes [189] such as vanillin [190], salicylc aldehyde [191] and cinnamaldehyde [192] were proposed (Figure 10).

The choice of a proper aldehyde determines the self-assembling behavior of the system [193].

## 5. Chitosan-Based Organic–Inorganic Hybrids for Biomedical Applications

Based on the IUPAC definition, a hybrid material is a “material composed of an intimate mixture of inorganic components, organic components or both types of components, where the components usually interpenetrate on scales of less than 1 μm” [173]. This definition embraces organic-inorganic hybrids containing at least one inorganic component and one organic component with an interpenetrating structure on the sub-micrometric scale. Due to their dual composition, organic–inorganic hybrids display unique properties, offering interesting opportunities in several applications [194], including in regenerative medicine [195]. The interpenetrating network formed by the organic and inorganic matrices allows finely tuning the mechanical and biological properties, combining the high mechanical properties typical of the inorganic components, with the elasticity [196], and possibly biocompatibility and bioactivity of the organic components [197,198,199,200]. Additionally, a suitable hybrid formulation allows obtaining organic–inorganic hydrogels, monoliths, bulk materials and 3D printable inks [196].

The organic–inorganic hybrids can be further divided into two classes referred to as class I and II. In class I, the interconnected network is based on weak intermolecular forces, such as Van der Waals, hydrogen bonds, ionic and electrostatic interactions. The class II hybrids are featured with stable covalent bonds between the organic and inorganic phases. Obviously, in class II hybrids coexistence of covalent and non-covalent interactions is possible. The exploitation of CHT as the organic component is particularly attractive [201] due to its biocompatibility, biodegradability, low toxicity and biological properties, as previously highlighted. At the same time, silica and hydroxyapatites are very appealing inorganic phases to be used in hybrid synthesis for regenerative medicine, especially for bone tissue repair. Both hydroxyapatite, the native bone mineral phase, and silica are pivotal to improve the mechanical properties of the final material, besides being osteoinductive and biocompatible.

### 5.1. Class I Chitosan-Based Hybrids

As in the case of hydrogel design, the polycationic nature of CHT at acidic pH, or after stable *N*-peralkylation, offers an easy way to form interpenetrating organic–inorganic hybrids with anionic counterparts. A robust network produced by ionic interactions between the protonated amino groups of CHT and hydroxysilicate anions was obtained by a sol-gel process, performed in acidic conditions, by mixing CHT and hydrolyzed tetraethoxysilane (TEOS) solutions [202]. The role of pH and the acidic nature was investigated. It was found that hybrid materials with better mechanical properties could be obtained in mild acidic conditions (pH 4) in the presence of acetic acid. The modulation of the silica/CHT ratio also allowed tuning the mechanical properties: the higher the CHT content, the lower the rigidity of the resulting scaffold. The bioactivity was investigated in terms of bone-like hydroxyapatite deposition on the scaffold surface. Notably, scaffolds with CHT content higher than 50% did not trigger the formation of hydroxyapatite, thus limiting the osteoinduction properties of the silica phase.

Hydroxyapatite scaffolds can be prepared with suitable porosity and morphology. However, their mechanical properties lack the characteristic elasticity of human bone tissue. Additionally, to improve the osteogenic potential of the scaffolds, bioactive factors such as bone morphogenetic proteins are often included [203,204]. However, the biological macromolecules are often expensive, thus limiting clinical applications. To circumvent these limitations, hybrids with an organic component have been proposed: class I hybrids were prepared using CHT and hydroxyapatite doped with therapeutic metal ions, including copper (II) and strontium [205], as a low-cost and effective alternative to osteogenic macromolecules [206]. Strontium was considered due to its ability to promote osteogenesis and bone remodeling, while copper promotes angiogenesis and denotes antibacterial activity. Preliminary in vitro biological evaluation of MG-63 human osteoblast-like cell viability highlighted promising features of these hybrids for bone tissue regeneration (Figure 11). However, further studies are needed to prove their potential.

### 5.2. Class II Chitosan-Based Hybrids

Covalent bonds between the organic and inorganic phases in class II hybrids bring similar advantages when compared to physical interactions, as previously reported for hydrogels. In class II hybrid synthesis, the covalent bonds are commonly obtained by means of suitable cross-linking agents, possessing both a reactive moiety towards the inorganic component, and one with complementary reactivity to the organic (macro)molecule [207]. Within this framework, functional organosilanes, such as 3-glycidoxy propyl trimethoxysilane (GPTMS) [208,209] and aminopropyl triethoxysilane (APTES) have been widely used as cross-linking agents. For example, GPTMS has been used for the synthesis of hybrid scaffold with oriented structures through the sol–gel methodology, followed by a unidirectional freeze-casting step, affording a highly ordered lamellar pore structure (Figure 12a–c). The covalent bond between the inorganic silica network and organic CHT phase was obtained by the nucleophilic attack of the CHT amino group on the epoxide of the GPTMS moiety (Figure 13), followed by silica network formation.

However, the conjugation of CHT to GPTMS has been reported to occur in very low yields due to the competing water nucleophilic attack, leading to the opening of the epoxide to the corresponding diol. However, the CHT-silane conjugate enabled obtaining covalent hybrids, and their biomechanical properties could be tuned by varying the ratio of organic/inorganic components. In particular, when the organic component was the major component (60%), the resulting scaffolds showed flexible and elastomeric behavior perpendicular to the freezing direction, while having elastic-brittle features parallel to the freezing direction (Figure 12) [210]. This anisotropic mechanical response, which is intrinsically dependent on the direction of stress, is similar to what occurs in cartilage, where the orientation of collagen fibrils depends on the articulation zone.

Sol-gel chemistry is usually based on tetraethoxysilane (TEOS) as the precursor for the inorganic phase, which upon condensation releases ethanol as the reaction by-product, that is a cytotoxic compound that inhibits cell growth and proliferation. The less toxic glycerol-modified silane precursor was proposed for the synthesis of novel CHT-containing class II hybrids, denoting a hydrogel behavior [211]. As in the previous example, the cross-linker was GPTMS; however, a mixture of CHT and a thiolated-CHT was used in the conjugation reaction. In preliminary biological assays, the hybrids showed to be cytocompatible, and revealed antibacterial activity and suitable drug delivery properties.

## 6. Chitosan-Based Layer-by-Layer Assemblies for Biomedical Applications

The layer-by-layer (LbL) assembly technology is an easier, cost-effective and highly versatile bottom-up methodology to readily and conformally coat surfaces and develop a wide array of multilayered devices with finely tuned properties and functions at the nanoscale. The technology simply relies on the sequential adsorption of at least two distinct building blocks exhibiting complementary intermolecular interactions on virtually any type of surface, leading to a diverse set of multilayered structures [212].

Dating back to the early works by Iler on the dip-assisted LbL assembly of oppositely charged colloidal particles on flat glass substrates in 1966 [213], and Decher and Hong on either oppositely charged bipolar amphiphiles [214] or polyelectrolyte multilayers [215], and combinations thereof [216] on charged planar surfaces in the early 1990s, the electrostatic interaction between oppositely charged materials is still the most employed build-up mechanism of multilayered assemblies. Moreover, although the dipping methodology has been by far the most used methodology owing to its feasibility in coating substrates of any size, shape or surface chemistry, the need for a large amount of materials and its time-consuming process turned attention to other processing methodologies, including the commonly employed spin-coating and spraying [217,218,219]. However, over the last two decades other fabrication methodologies have emerged such as fluidic- and electromagnetic-driven assembly [218], high-gravity field- and inkjet printing-assisted assembly [220,221,222,223,224], and LbL assembly on particles [225,226], thus opening new avenues for which the technology can be applicable [218].

The fact that the LbL assembly process can be performed under mild conditions in entirely aqueous solutions enables assembling biological molecules (e.g., proteins, enzymes, polysaccharides) and cells, preserving their biological activity, thus being highly advantageous in addressing biomedical and biotechnological applications [227,228,229,230,231,232,233,234,235]. However, the versatility imparted by the LbL assembly technology expands well-beyond the use of biological molecules to enable the adsorption of a wide array of constituents (e.g., nanoparticles, synthetic polymers, clays, carbon nanotubes, dyes, metal oxides) on virtually any type of substrate, regardless of size, shape, surface chemistry, and even animate or inanimate nature towards shaping multifunctional multilayered devices across multiple scale lengths [236].

Herein, we emphasize the combination of chitosan biopolymer with either other natural or synthetic ingredients to shape a wide array of LbL structures, including nanostructured multilayered thin films and thicker free-standing membranes, multilayered particles, hollow capsules and (multi)compartmentalized systems, hollow tubes, scaffolds/constructs, and animate living cell surfaces for addressing biomedical applications.

### 6.1. Multilayered Thin Films and Free-Standing Multilayered Membranes

Since its early-stage development, the LbL assembly technology has been widely applied to coat a wide variety of 2D flat hydrophilic and hydrophobic non-patterned substrates, ranging from glass, quartz, polystyrene, silicon wafer, and gold to produce 2D nanostructured multilayered thin coatings for biomedical and biotechnological applications. Although tightly bound to the underlying substrate, chitosan-derived nanostructured multilayered thin films have revealed to be very appealing nanocoatings for the loading, protection, transport, and on-demand controlled release of cargo, ranging from small molecules to large macromolecules, as well as to control cell functions. For instance, positively charged chitosan was combined with oppositely charged alginate (ALG) into spray-assisted ALG/CHT wholly marine polysaccharide-derived multilayered coatings on polyethyleneimine(PEI)-functionalized glass substrate for the loading and controlled release of tamoxifen (TMX), a well-known breast cancer drug [237]. It was shown that the release profile of the drug physically adsorbed onto the PEI/(ALG/CHT)_5_ base layer could be modulated by playing with the number of (CHT/ALG)*_n_* bilayers from 5 to 20, being faster for the nanocoating with the lowest number of bilayers. Moreover, irrespective of the number of bilayers, the TMX-loaded multilayers proved to be efficient therapeutic nanocoatings in reducing the MCF-7 human breast cancer cells viability in vitro, holding potential to be used as patches for sustained TMX release. Similar biocompatible (ALG/CHT)_5_ multilayered thin films were assembled on PEI-coated polystyrene cell culture plates and further cross-linked with genipin (G) towards modulating human umbilical vein endothelial cells’ (HUVECs) functions [238]. It was found that the cross-linked nanocoatings enhanced cell adhesion, spreading and proliferation in both normal and serum-free medium when compared with the uncross-linked counterparts owing to their higher stiffness and lower hydration level (Figure 13). Adding to this, the addition of an extra CHT/ALG bilayer over the cross-linked nanofilm reverted the cell adhesion, spreading and proliferation to similar levels as those obtained in the uncross-linked (ALG/CHT)_5_ nanofilm, mainly in the case of those cultured in serum-free conditions, thus revealing the key role of the nanocoating stiffness and surface properties in tuning both cell-nanocoating interactions and cell behavior. Quartz has been also widely employed as a template for assembling multilayered nanofilms for sustained drug release and to control cell behavior. For instance, in a recent study four CHT/hyaluronic acid (CHT/HA) bilayers were assembled as base layers into a quartz substrate followed by the adsorption of (CHT/HA-siRNA)*_n_* bilayers (*n* = 0–15 bilayers), as monitored by UV-visible spectroscopy. This study revealed that the multilayers were only effective in enabling the controlled release of siRNA from the nanofilms when assembling up to nine siRNA-loaded bilayers. Furthermore, the siRNA-loaded multilayered thin films promoted good cell adhesion and siRNA silencing effect in enhanced green fluorescent protein (eGFP)-HEK 293T cells, as demonstrated by the decrease in the eGFP expression. These surface-mediated non-viral multilayered nanofilms hold great promise as nanoplatforms for site-specific sustained release of siRNA in mucosal tissues [239]. Besides, silicon wafer, titanium and gold have also been among the most widely used flat substrates for assembling either wholly marine polysaccharide-based or hybrid multilayered nanofilms with tunable physicochemical and biological properties and multifunctionalities for a variety of bioapplications [240,241,242,243,244,245,246,247,248]. Notably, the limitation imposed by the need for assembling CHT-based multilayered thin films under slightly acidic pH (pH < pKa ≃ 6.2) has been recently surpassed by the synthesis and further assembly of quaternized CHT (Q-CHT))/heparin multilayers under physiological conditions, holding great promise as nanoreservoirs of proteins to control cell functions under non-denaturing conditions [249]. Such an approach could be virtually translated into the assembly of Q-CHT with any oppositely charged material into multilayer films.

More recently, considerable attention has been devoted to the assembly of chitosan-derived multilayered thin nanofilms onto flat patterned substrates for biomedical purposes owing to the possibility to impart the functional multilayered coatings with topographical features reminiscent of the substrate properties. In this regard, polystyrene superhydrophobic surfaces decorated with patterned wettable regions of tunable size and geometry developed through bench-top approaches were used to build-up arrays of patterned and adhesive LbL films encompassing chitosan and oppositely charged dopamine-functionalized hyaluronic acid (HA-DN), denoting a different number of catechol groups via high-throughput screening (Figure 14a) [250]. The in vitro cellular assays and mechanical tests revealed that the multilayered nanofilms having the larger amount of dopamine conjugated to hyaluronic acid (HA-4DN) showcased an enhanced cell adhesion and highest adhesive strength, which increased upon increasing the number of CHT/HA4-DN bilayers (Figure 14b,c).

Although multilayered thin nanofilms cannot be easily detached from the underlying substrate and are not robust enough to be used in practical biomedical applications, they provide important information about the feasibility and growth mode, as well as of the structure and properties of the multilayered films. Such knowledge has been translated into the assembly of robust and thicker free-standing multilayered membranes encompassing a huge number of layers, which are much more prone to be translated into innovative devices to fulfill biomedical purposes. Thicker multilayered films have been widely assembled on either hydrophobic/hydrophilic non-patterned or patterned substrates by resorting to an automatic dipping robot and detached into free-standing multilayered membranes. The chosen type of substrates, namely its topography, dictates the final end-use of the as-produced free-standing membranes whose topography is reminiscent of the substrate’s features. Hydrophobic, inert and low surface energy non-patterned polypropylene (PP) substrates have been among the most used templates to produce readily detachable free-standing membranes, since there is no need for either harmful solvents, extreme temperatures or sacrificial layers to detach the assembled multilayered films from the underlying hydrophobic template. In fact, upon reaching a certain number of bilayers, which is dependent on the chosen material’s combinations, the as-produced multilayered assemblies can be easily detached into free-standing membranes using solely tweezers. In this regard, chitosan has been combined with a diverse set of oppositely charged natural biopolymers, including ALG, chondroitin sulfate (CS) or hyaluronic acid to produce free-standing membranes with tunable properties and functions that could be applied in numerous biomedical applications [251,252,253]. For instance, CHT/ALG free-standing membranes were crosslinked with genipin to produce membranes with higher stiffness and better cell adhesion for tissue engineering strategies [253]. Similar CHT/ALG free-standing membranes but with gradients of increasing stiffness were produced by continuously increasing the level of genipin-induced cross-linking of the free-standing membranes along the time upon exposure to a solution with increasing levels of genipin [254]. It was found that the membranes with high cross-linking degree showcased enhanced mechanical properties and better cell adhesion and proliferation when compared with the membranes with low cross-linking degree or even the native ones (Figure 15). Such cross-linked membranes also showed to be suitable reservoirs of biomolecules, including growth factors holding great potential for being used as wound dressing devices [255].

Native and genipin-induced cross-linking free-standing membranes made of CHT/ALG multilayers also exhibited shape memory properties, undergoing reversible shape switching triggered by either hydration or ionic cross-linking [256,257]. Moreover, the incorporation of magnetic nanoparticles in such self-standing CHT/ALG membranes imparted them with shape memory and magneto-responsive properties, which improved cell adhesion [258]. Those membranes hold invaluable potential as smart implantable devices to be inserted in the human body in a temporary shape via minimally invasive procedures, and once reaching to the injured site and achieving a certain hydration level would adopt its permanent shape. Biomimetic mussel-inspired multilayered membranes denoting tunable and improved adhesive and mechanical properties, as well as cell adhesion and proliferation, were also produced on smooth polypropylene substrates by electrostatically assembling either 200 CHT/HA-DN bilayers [259], or integrating HA-DN in the assembly of 100 CHT/ALG/CHT/HA-DN tetralayers [260]. The in vitro biological performance of the as-produced membranes was assessed with different cell types, including human primary dermal fibroblasts and MC3T3-E1, demonstrating their intrinsic potential in skin wound healing and bone tissue engineering, respectively. Moreover, the incorporation of both HA-DN and bioactive glass nanoparticles (BGNPs) into hybrid nacre-inspired bioresorbable free-standing membranes containing CHT/HA-DN/CHT/BGNPs tetralayers rendered them not only bioadhesive but also bioactive, as showcased by the formation of a calcium-phosphate layer on their surface [261]. As such, the membranes denote immense potential to act as wound dressings, as well as bioinstructive matrices to promote guided bone tissue regeneration in addressing periodontal diseases.

Self-standing membranes have also been assembled on patterned hydrophobic templates aiming to better control cell functions in addressing specific tissue engineering strategies. For instance, highly aligned tissues such as muscle, blood vessels or nerves would undoubtedly benefit from the assembly of free-standing membranes whose nanoscale topographical features would be reminiscent of those of such tissues. In fact, patterned polycarbonate templates exhibiting nano-grooved features were used to produce robust self-standing nanopatterned cross-linked chitosan-chondroitin sulfate (CHT/CS) membranes that directed C2C12 myoblast cell alignment along the nanopattern direction and triggered their differentiation into myotubes when using non-differentiated growth medium, thus being promising bioinstructive matrices for muscle regeneration (Figure 16) [262]. Such a platform could be adapted to other cell types existing in highly aligned tissues, such as neuronal or endothelial cells to enable the regeneration of nerve tissue or blood vessels, respectively. Furthermore, patterned polydimethylsiloxane templates denoting an array of micro-wells with tunable geometry were designed to precisely assemble micro-pore mimetic free-standing CHT/ALG membranes in which human osteoblast-like cells tended to colonize preferentially [263]. Those membranes featuring micro-pores could be used as micro-reservoirs for the loading, protection, and on-demand sustained release of bioactive molecules or as cell carriers in regenerative medicine strategies.

### 6.2. Multilayered Particles, Hollow Multilayered Capsules and Hierarchical (multi)Compartmentalized Capsules

The versatility imparted by the LbL assembly technology has been well-demonstrated by its potential to coat more convoluted 3D surfaces, including colloidal particles, tube-like, hierarchical multi-compartmentalized or porous structures, thus extending its applications in the biomedical arena. Organic and inorganic biocompatible templates have been widely used to prepare core-shell multilayered particles. Such particles are engineered by repeating the alternate and sequential adsorption of aqueous solutions of complementary materials onto the particles’ surface. The use of sacrificial core templates further enables the preparation of hollow multilayered microcapsules following core template dissolution.

Microsized calcium carbonate (CaCO_3_) inorganic particles have been widely employed as templates for the fabrication of core-shell LbL micro-particles and hollow microcapsules for bioapplications owing to their unique features, including easy, fast and inexpensive synthesis, highly porous structure, large surface area-to-volume ratio, biocompatibility, non-toxicity, and high mechanical stability. Chitosan has been combined with other oppositely charged natural [264], (Figure 17) or synthetic polymers [265,266] to engineer fully natural or biomimetic chitosan-derived multilayered shells, respectively, templated on spherical CaCO_3_ microcores. The exposure of the sacrificial template to calcium-chelating agents such as ethylenediamine-tetraacetic acid (EDTA) enabled its decomposition and development of hollow multilayered microcapsules (Figure 17b), which are particularly attractive as multifunctional carrier vehicles of high payloads for intracellular delivery, cellular internalization (Figure 17c) or intracellular trafficking.

Organic biocompatible and biodegradable polymers have also been employed as templates to assemble chitosan-based core-shell multilayered micro- and macroparticles and further liquefied multilayered capsules to be used for cell encapsulation in in vitro and in vivo tissue engineering and regenerative medicine strategies [267]. In particular, hybrid multilayered nanoshells encompassing oppositely charged poly(L-lysine) (PLL), ALG and CHT were templated on spherical calcium chloride cross-linked alginate micro- and macroparticles loaded with either a mono- or co-culture of cells and surface functionalized poly(*L*-lactic) (PLA) or poly(ε-caprolactone) (PCL) microparticles, providing anchorage sites for enabling cell adhesion and proliferation [225,226,267,268,269,270]. The exposure of the core-shell particles to EDTA induced the chelation of the calcium ions, producing liquefied alginate multilayered micro- and macrocapsules featuring a chitosan-derived LbL shell. Such shell revealed to be permselective, enabling the inwards diffusion of nutrients and oxygen essential to sustain cell survival and the excretion of cell metabolites and waste products, while excluding the entrance of large molecules of the host immune system and other cells. On the other hand, the liquefied core maximizes the diffusion of those essential molecules along the entire system, thus surpassing the limitations of larger-size tissue constructs (above ca. 200 μm). The multilayered capsules could be cultured under dynamic tissue-like conditions and encapsulate virtually any biomolecules of interest and anchorage-dependent cell types, thus enabling the in vitro development of microtissues in a more close-to-native and inexpensive manner. In particular, the liquefied and multilayered microcapsules have shown to encapsulate a co-culture of human adipose-derived stem cells (hASCs) and either human adipose-derived microvascular endothelial cells or osteoblasts anchored to collagen I-functionalized PLA or PCL microparticles and to form bone-like microtissues in vitro and in vivo, holding great promise to be applied in bone tissue engineering [267,268,270]. Besides, similar liquefied microcapsules encapsulating supportive PCL microparticles and a mono- or co-culture of HUVECs and human fibroblasts have been produced with a bioactive multilayered nanoshell encompassing PLL/ALG/CHT and an outer alginate layer functionalized with arginine-glycine-aspartic acid (RGD) tripeptide cell adhesive motif to enhance their biological performance [271]. 3D microaggregates of cells and microparticles were produced and confined within the liquefied core by cells recruiting microparticles and 3D microcapsule macroaggregates were formed in the outside by cells and deposited extracellular matrix, whose linkage is promoted by the outermost RGD bioactive peptide-functionalized microcapsules. The bioactive outer layer promoted the recruitment of new microvessels and formation of vasculature, thus enabling the diffusion of essential molecules for cell survival and possibly stimulating a proper integration of the microcapsules within the surrounding tissue in vivo to better foster tissue regeneration. Magnetic-responsive liquefied alginate macrocapsules denoting an LbL nanoshell encompassing oppositely charged PLL, ALG, CHT and magnetic nanoparticles (MNPs) have shown to encapsulate hASCs cells anchored to crosslinked collagen II/TGF-β3–surface functionalized PLA microparticles and induce their chondrogenic differentiation aiming at cartilage tissue regeneration. Those self-regulated liquefied multilayered microcapsules have great potential to be applied as innovative bioencapsulation systems and platforms for multiple tissue engineering and regenerative medicine strategies.

Liquefied spherical alginate macrocapsules functionalized with a tunable CHT/ALG LbL nanoshell have also been used as reservoirs of model fluorophores and CaCO_3_-templated temperature-responsive chitosan/elastin-like recombinamers (ELRs) multilayered microcapsules, thus enabling multifunctional, (multi)compartmentalized capsules with a hierarchical organization from the nano- to the macro-scale mimicking living systems [272]. The latter internal microcompartment further encapsulated either magnetic nanoparticles (MNPs) or fluorescent model molecules, whose release could be tailored and spatiotemporally controlled on-demand by playing with the temperature-sensitive nature of ELRs and magnetic field-responsive MNPs, respectively. Such multicompartmentalized capsules could encapsulate virtually any type of compartments, including multistimuli-responsive ones, thus enabling smart multifunctional systems that would be very appealing for a wide variety of biomedical and biotechnological applications.

### 6.3. Hollow Multilayered Tubes

Cylindrical substrates have also been coated in an LbL fashion to engineer innovative self-sustained hollow multilayered tubes, with tunable properties and functions at the nanoscale, which hold great promise in sustained drug/therapeutics delivery, tissue engineering and regenerative medicine. For instance, self-standing hollow multilayered macrotubes (*ca.* 1 mm) were engineered by dip-coating sacrificial paraffin wax-coated glass tubes in (ALG/CHT)_100_ multilayered nanoshells followed by dichloromethane-induced core template leaching, without altering the nanoshell properties [273]. Owing to their softness and high hydration level, the native hollow tubes were further crosslinked with the natural cross-linking agent genipin, leading to tunable multilayered tubes with decreased water uptake, increased stiffness and improved L929 fibroblast cells adhesion and proliferation when compared with the native ones. Such a proof-of-concept study launched the seeds for a follow-up work aiming to develop tube-like artificial blood vessel substitutes for cardiovascular tissue engineering. In this regard, similar genipin cross-linked ALG/CHT hollow multilayered tubes were bioengineered and subsequently functionalized with fibronectin via EDC/NHS chemistry to further enhance cell adhesion (Figure 18a,b) [274]. The as-produced hollow tubes were successfully co-cultured with human umbilical vein endothelial cells in the inner side and human aortic smooth muscle cells in the outer side, two cell types existing in the blood vessels’ composition, thus recreating the native blood vessels (Figure 18c). Besides, (ALG/CHT)_8_ hollow multilayered nanotubes templated on the inner pores of PEI-functionalized polycarbonate templates (Figure 18d,e) have shown to be internalized by cancer cells (Figure 18f), holding great potential as carrier vehicles of therapeutic agents [275].

The versatility imparted by the hollow multilayered tubes would enable their use as reservoirs of bioactive agents to enable long-term cell survival and trigger the formation of pre-vascularized tissues. Moreover, owing to their geometry and tunable properties and multifunctionalities at the nanoscale, the hollow multilayered tubes hold great promise as reservoirs of neuronal growth factors and neuronal cells for repairing neuronal networks in neural tissue regeneration, as well as platforms to enable the development of (multi)compartmentalized systems.

### 6.4. 3D Constructs

In the last decade, LbL technology has been moving a step forward to assemble even more complex 3D structures for use in the biomedical and biotechnological fields. Spherical paraffin wax particles were surface functionalized with PEI prior to their conformal ALG/CHT multilayered coating via the perfusion LbL assembly methodology, using a perforated cylindrical container, thus leading to 3D interconnected cylindrical structures encompassing multilayered microspheres denoting a regular stacking arrangement (Figure 19a(i)) [276]. The paraffin core was leached out by exposure to dichloromethane, enabling moldable 3D interconnected and porous self-supporting multilayered constructs (Figure 19a(ii)). The beneficial effect of both the LbL coating and particles’ interconnectivity on cell adhesion and viability was studied using human osteoblast-like cell lines, revealing that more than 99% of cells seeded on the construct remained viable and metabolically active after 3 days of culture through the entire structure (Figure 19a(iii)). The perfusion LbL assembly methodology and spherical template leaching have also been combined to coat 3D packet paraffin spheres with chitosan/chondroitin sulfate (CHT/CS) multilayered nanoshells, which were further leached out to assemble highly porous and interconnected nanostructured 3D CHT/CS multilayered constructs [277]. Such constructs proved to support the adhesion, proliferation, and viability of either bovine chondrocytes or multipotent bone marrow-derived stromal cells and the chondrogenic differentiation of the latter, thus holding potential to be applied in cartilage repair. This approach can be adapted to virtually any template, irrespective of the size and geometry, and a multitude of complementary LbL biopolymers can be assembled into multilayered films, which could act as a reservoir of bioactive molecules. Furthermore, virtually any type of cells can be cultured, thus opening new horizons in a wide array of tissue engineering and regenerative medicine strategies. More recently, packed calcium cross-linked cell-laden ALG beads or reeled fibers with tunable size and geometry were conformally coated with CHT/ALG multilayers via perfusion-based LbL methodology and further chelated with EDTA to produce 3D self-standing liquefied constructs (Figure 19b(i,ii)) [278,279]. Such modular constructs were revealed to be well interconnected and could be easily handled without disrupting their original 3D structure by the action of the assembled multilayers. Moreover, the assembled liquefied 3D constructs enabled the survival and proliferation of L929 fibroblast cells (Figure 19b(iii)). Additionally, the cell proliferated freely throughout the entire liquefied system, including inside the liquefied bound beads or fibers, thus revealing their cytocompatibility and permeability of the multilayered coating, which is key to regulating the diffusion of oxygen, nutrients, and metabolic waste products to ensure cell survival.

Porous 3D hierarchical micro- and macro-scaffolds also were prepared in a straightforward way by combining the LbL assembly technology with the rapid prototyping technique. In this regard, prototyped 3D PCL macro-scaffolds were coated in an LbL fashion with bioinstructive nanocoatings encompassing oppositely charged marine-origin polysaccharides, namely CHT and carrageenan, and human platelet lysate towards assembling hierarchical cell-instructive 3D multiscale scaffolds [280]. The LbL-coated scaffolds were further freeze-dried to shape the bioinstructive nanoassemblies in the inner side of the scaffolds into fibrillar structures, which provided enhanced cell-anchorage points to induce the osteogenic differentiation of hASCs into osteoblasts. The proposed methodology could be translated into the assembly of any bioactive LbL coatings, which can act as reservoirs of bioactive molecules, and different cell types can be seeded to direct multiple tissue engineering purposes.

Other 3D scaffolds were prepared by resorting to alternative methodologies and further LbL-coated to impart the scaffolds with enhanced properties and multifunctionalities for being used in biomedical applications [281,282]. A recent original research article combined 3D printing with the LbL assembly technology to bioengineer customized large 3D constructs (Figure 20a) aimed at surpassing the bulk hydrogels-limited diffusion of oxygen and nutrients essential to sustain long-term cell survival and enable the formation of prevascular networks for vascular tissue engineering [283]. Customizable calcium cross-linked sacrificial ALG structures exhibiting tunable sizes and shapes (Figure 20b), including microfibers were 3D printed, coated with six bioinstructive chitosan/RGD-grafted alginate multilayers in an LbL fashion and further embedded in a shear-thinning and bioinert photocrosslinkable xanthan gum hydrogel (XG-GMA). The immersion of the full 3D construct in EDTA induced the liquefication of the alginate core template, leading to perfusable bioinstructive hollow multilayered microstructures, which were fixed and sustained without collapsing by the supporting hydrogel matrix (Figure 20c). HUVECs were seeded in the inner walls of the bioinstructive LbL-coated perfusable microchannels with FBS-free culture medium, revealing a much higher number of adherent and viable cells when compared with the uncoated microchannel (90% vs. 5%, Figure 20d). The bioinstructive LbL-coated microchannels embedded in hydrogels hold great promise to bioengineer endothelial cell-lined tubular networks as blood vessel substitutes. Moreover, the versatility imparted by the combination of 3D printing, LbL assembly technology and photocrosslinkable hydrogels open new perspectives in bioengineering large-scale 3D vascularized tissue constructs for modular tissue engineering and regenerative medicine.

### 6.5. Living Cell Surfaces

The use of entire aqueous solutions and the mild processing conditions behind the fabrication of the LbL nanoassemblies have turned this technology into a suitable, cytocompatible methodology to functionalize animate and dynamic living cell surfaces, including single cells and cell aggregates [284,285,286,287,288]. In fact, probiotic microorganisms have been encapsulated in CHT/ALG multilayers to protect them from the gastro-intestinal microbiome and enhance their delivery, adhesion and growth in vivo (Figure 21a) [289]. Furthermore, the chitosan-based LbL nanocoating improved not only the probiotic viability, but also facilitated its mucoadhesion and growth on the porcine intestine surface when compared with the uncoated probiotic (Figure 21b).

In other studies, the encapsulation of the bacteria *Escherichia coli* or *Staphylococcus epidermidis* in biopolymeric nanoshells of chitosan and either alginate or dextran sulfate preserved the viability and delayed the growth of the encapsulated bacteria when compared with uncoated bacteria [290,291]. Such a strategy represents a promising approach to prevent the proliferation of both bacteria and, thus, maintain the gut or skin microbiota aiming to regulate human health. The LbL technology was also used as an efficient way to enhance and modulate live *Bacille Calmette-Guérin* (BCG) mycobacteria’s immunogenicity by functionalizing its surface with a polymeric nanocoating containing oppositely charged chitosan and strong immunostimulatory agent polyinosinic–polycytidylic acid (poly(I:C)), a synthetic analog of the double-stranded RNA [292]. It was found that the multilayered nanocoating induced a stronger and long-term protective immune response against adult pulmonary tuberculosis. The multilayered nanocoating did not affect the bacterial viability and further induced an enhanced macrophage pro-inflammatory response and expression of co-stimulatory molecules when compared with the uncoated BCG.

## 7. Chitosan-Based Inks for 3D Printing Applications

In the last decades, regenerative medicine and tissue regeneration may include bioabrication strategies, defined as “the automated generation of biologically functional products with structural organization from living cells, bioactive molecules, biomaterials, cell aggregates such as micro-tissues, or hybrid cell-material constructs, through bioprinting or bioassembly and subsequent tissue maturation processes” [293]. Within this framework, 3D printing, an additive manufacturing technology consisting of the LbL fabrication of 3D scaffolds with tunable sizes and geometries, programmed by means of computer-aided drafting (CAD), plays a relevant role as a promising tool for the replacement of damaged tissues. Differently from the off-of-shelf scaffold production or the common hydrogels scaffolds, 3D printing may allow the fabrication of novel 3D bioengineered tissue with promising properties [294,295].

When aiming for biomedical applications, the accurate choice of the ink, namely the development of a printing formulation with proper rheological properties that may contain bioactive (macro)molecules and biomaterials, allowing spatial organization and cell growth, is vital. In this regard, of particular interest are bioinks (Figure 22), “formulations of cells suitable for processing by an automated biofabrication technology that may also contain biologically active components and biomaterials” [296].

Designing bioinks with suitable printing properties is still a difficult task. Notably, CHT can be exploited as a (bio)ink component due to its low cost, biocompatibility and non-immunogenicity. However, its use as a component in 3D bioprinting applications is reported roughly in 4% of publications. In this section, the recent developments in the field of 3D (bio)printing are highlighted, with a focus on the use of CHT as a (bio)ink component aiming at maximizing key parameters: the printability of the (bio)ink, cell viability after the printing step for regenerative applications and loading/release ability as drug delivery platforms [297,298].

The possibility to locally deliver drugs/therapeutics with printed scaffolds encompassing inks of varied composition has been reported for breast cancer treatment, including using a polycaprolactone/CHT ink. CHT, itself, is not suited for 3D printing due to its poor ductility and mechanical strength. However, it is well suited as a drug delivery platform, as described in the previous sections. On the other hand, PCL, due to its glass transition temperature and in vivo biodegradability, possess suitable features for 3D printing and biocompatible scaffold design. However, it lacks favorable interactions with cells and tissues, due to its hydrophobicity. Hence, the combination of CHT and PCL, combining their respective features, may encompass limitations of each single polymer. A two-layer 3D printed scaffold was designed: in one layer, PCL is coupled with CHT for the release of the drug (5-Fluorouracil); the second layer encompass PCL, where gold nanoparticles (AuNPs) were subsequently loaded. AuNPs may act as radiation enhancers or local heat generators upon infrared irradiation, thus contributing to cancer cells’ death by local temperature enhancement. The PCL/AuNPs-PCL/CHT printed scaffold maintained a flexible structure suitable for the implant in the human body (Figure 23), showed a good drug release profile and antibacterial activity due to the CHT properties, and long scaffold degradation time owing to the PCL component, useful for the drug release profile [299].

Skin tissue engineering is an emerging field where 3D bioprinted constructs have huge potential in applications due to their ability to be used as reservoirs of drugs and in situ wound healing platforms. The long-term efficacy of scaffolds denoting long-term antibacterial activity and high biocompatibility mimicking the native epithelial ECM environment are highly desirable for this application. Towards this aim, an ink composed by a soluble quaternized CHT derivative (Figure 24a), gelatin, and decellularized ECM (dECM) derived from fresh porcine skin was designed. However, since the resulting scaffolds lacked thermostability at a physiological temperature, a cross-linking step via EDC/NHS coupling chemistry was performed. Finally, to impart antibacterial properties, a polycationic polymer (Figure 24b) was incorporated within the scaffold composition.

The resulting 3D printed scaffold (Figure 25) was studied as a wound healing platform for skin tissue regeneration in terms of hemostatic and antimicrobial activities, cell biocompatibility, adhesion and proliferation induction. The scaffold showed 100% of antibacterial activity against *E. coli* (Gram-negative) and *S. aureus* (Gram- negative) bacteria, and good hemostatic and hemocompatibility properties when compared with a quaternized CHT-free control scaffold. Additionally, it allowed the growth of fibroblasts, showing a higher ECM deposition, in particular, in terms of collagen type I, fibronectin and decorin [300].

An interesting approach towards bone tissue regeneration is provided by a bioink obtained by inducing a fast gelation (about 7 s at 37 °C) of different components, namely CHT, glycerophosphate, hydroxyethyl cellulose and cellulose nanocrystals. This bioink is printed in the presence of the pre-osteoblast lineage MC3T3-E1 cells. The presence of both cellulose nanocrystals and cells were key factors in determining the rheological properties of the bioink, and improving the mechanical features of the final CHT-based scaffold. Notably, a wide range of printing pressures could be used (12–20 kPa), without having a detrimental effect on cell viability. Increased alkaline phosphatase activity, a marker of osteogenesis, together with cell differentiation into osteoblasts, calcium phosphate nucleation and ECM deposition were observed, highlighting that CHT and cellulose nanocrystals-based bioinks could be considered as a valid approach for bone repair [301].

CHT can be functionalized with acrylic moieties, which upon photopolymerization reaction afford hydrogels suitable for bioprinting. A recent study proposed a bioink obtained by the combination of keratin and glycol CHT methacrylate, which are polymerized by UV-mediated photo-crosslinking. The produced bioink is an example of a tunable ECM mimicking material, promoting cell growth and adhesion. The optimization of the bioink composition allowed bioprinting monodispersed cells, as well as spheroids. The human adipose stem cell spheroids have been embedded in the keratin/CHT methacrylated bioink, showing a rapid migration of the cells through the matrix within five days from the cell encapsulation. In contrast to the spheroid’s derived cell behavior, cell suspension remained in their round morphology, supporting the hypothesis that cells in 3D architecture strengthen the cell-cell signaling [302].

Recent studies aimed to improve the printability conditions of CHT by formulating CHT-based bioinks that can be crosslinked by means of ionic cross-linkers in physiological conditions. One example is the use of nanohydroxyapatite, an inorganic phase widely exploited for bone tissue engineering applications, as emphasized in the previous sections. In particular, a CHT-based bioink cross-linked by glycerol phosphate and sodium hydrogen carbonate, in the presence of different ratios of nanohydroxyapatite has been proposed. The obtained 3D printed scaffolds have been analyzed in terms of morphology, rheological properties and shape fidelity to understand if the printing conditions were suitable for a robust biofabricated construct. The optimized hydroxyapatite/CHT bioink is suitable for printing with pre-osteoblastic MC3T3-E1 cells, supporting cell viability [303].

The chemistry of cross-linking may be detrimental to cell viability and growth; self-crosslinkable strategies have been studied to overcome this limitation. A recent example is an innovative self-crosslinkable ink, obtained by the conjugation of CHT to gallic acid, through the carbodiimide coupling chemistry. The CHT-gallamide derivative spontaneously undergoes oxidation at physiological pH, affording the corresponding *o*-quinones, which in turn react affording imine and 1,4-Michael addition adducts, providing the cross-links of chitosan chains (similarly to the reactions described in Figure 5). Its biocompatibility was demonstrated with NIH3T3 cell lines, which proved to be viable after 7 days of culture post-printing. The CHT-gallamide ink has an increased mechanical strength (about 337 kPa) that can be modulated by controlling the self-cross-linking process. Due to the mechanical properties, the ink can be printed in various complex geometrical shapes with a high fidelity to the CAD prototypes (Figure 26), being promising for tissue engineering applications [304].

Notably, bioprinting is not only associated with tissue engineering and regenerative medicine fields, but it can also be exploited to fabricate bioactive materials with complex geometries for immobilizing microorganisms; this approach is useful, for example, in downstream steps in microorganism-mediated biotechnological productions. Genipin is a widely explored agent to cross-link CHT (the chemistry of cross-linking is depicted in Figure 8). However, the kinetics of the conjugation reaction is too slow (in the order of hours) to guarantee the stability of the 3D printed construct. To circumvent this limitation, alumina and alginate have been considered as components of the bioink formulation. In this regard, CHT is first dissolved in 1% acetic acid, followed by the addition of powdered alumina, and lastly, alginate and bacteria. Genipin is added as the last component, right before the printing step. The genipin-mediated slow cross-linking reaction turned out to be an advantage, since no obstruction of the nozzle was observed. The rheological properties of the ink were modulated with different concentrations of alginate to modulate the scaffold’s rheological properties to the bacterial growth (*Escherichia coli)*. The observed antibacterial activity of CHT and/or genipin in control experiments of bioinks formulation without alginate was strongly reduced by the presence of the alginate. The geometry of the scaffold allows better circulation and availability of nutrients for the bacteria, sustaining their viability. This approach can be exploited to better grow bacteria in bioreactors [305].

In summary. finding the perfect match among optimized mechanical properties, printability and cell biocompatibility is the main challenge for the formulation of innovative and improved CHT-based (bio)inks. Owing to its properties, CHT is considered as an interesting biopolymeric material suitable for (bio)ink formulation in combination with other (bio)materials. Bioprinting is a successful technology that could take tissue engineering to the next level, and CHT can play a prominent role in defining a valuable class of bioinks for different applications.

## 8. Conclusions

From the previous sections, it clearly emerges that CHT can be exploited as a bio-based polymeric material for a huge number of applications, being included in different formulations, and manufactured towards varying structures with different compositions, morphologies, and geometries. An increasing number of research papers and patents have been published, and will continue to grow, fostered by valuable applications in regenerative medicine and following circular economy strategies. Sustainability issues will be key players in promoting CHT spreading in several industrial sectors and applications.

The increasing interest towards this polysaccharide will also drive researchers to address still existing limitations to its widespread application, such as the environmental impact caused by the extraction of chitin and its deacetylation to obtain CHT, the too high production costs to be competitive on the market with respect to other polymers, the solubility issues at physiological pH, and structural heterogeneity.

The environmental and production costs will be key issues for chitosan to enter the market as a competitive substitute of fossil-based feedstocks to produce innovative materials for several applications. In recent years, studies are emerging aimed at the detailed life cycle assessment (LCA) of chitosan production, specifically focusing on environmental and economic viability [306].

We are sure that in the next few years, CHT will strongly contribute to unprecedented advancements in the material sciences and biomedical fields, thanks to the progress in alleviating the key issues that are still a limitation to chitosan exploitation.

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
