# Peer review of "Chitosan-Based Biomaterials: Insights into Chemistry, Properties, Devices, and Their Biomedical Applications"

_marinedrugs, 2023, doi:10.3390/md21030147_

Round 1

Reviewer 1 Report

The review " Chitosan-based biomaterials: insights into chemistry, properties, devices and their biomedical applications" is very interressting  work and well presented revew. The structuration is good and it's well written. We can read it easily the paper, which is clear with appropriate references and figures.

The authors produced a pertinent work summirizing different potential applications of using Chitosan as biobased material which is a reel need nowadays, fo the indutrials and also for academics.

I have some comments to improve this work before publication:

It will be important to notice in this rewiew the environemental impact of using Chitosan, the process of Chitin extraction and desacetylation of chitin to obtain chitosan is still a limitation to explain that this polymer is not competitive on the market (the production costs was higlighted in the conclusions part but not the environmental aspect)

-  The abbreviation list should be homogenized (exple : DN, dopamine, instead of Dopamine, DN) the cited references should be homogenized as well

Author Response

Please find a point by point reply to reviewer comments.

REVIEWER COMMENT: It will be important to notice in this rewiew the environemental impact of using Chitosan, the process of Chitin extraction and desacetylation of chitin to obtain chitosan is still a limitation to explain that this polymer is not competitive on the market (the production costs was higlighted in the conclusions part but not the environmental aspect)
AUTHOR’S REPLY.  We appreciate reviewer suggestions towards the improvement of the conclusions section. A few lines have been added in the conclusions in order to highlights the points made by the reviewer and a relevant reference on the topic (ACS Omega 2021, 6, 36, 23038–23051 https://pubs.acs.org/doi/10.1021/acsomega.1c01672) has been added (lines 1537-1541):” The environmental and production costs will be key issues for chitosan to enter the market as competitive substitute of fossil-based feedstocks for the production of innovative materials for several applications. In recent years, studies are emerging aimed at the detailed life cycle assessment (LCA) of chitosan production, specifically focusing on environmental and economic viability [10].”
REVIEWER COMMENT:  The abbreviation list should be homogenized (exple : DN, dopamine, instead of Dopamine, DN) the cited references should be homogenized as well
AUTHOR’S REPLY. Abbreviation list has been revised according to the reviewer suggestion. References have been edited thanks to zotero reference manager, following the Marine Drugs style.

Reviewer 2 Report

Review of Manuscript for Marine Drugs

Manuscript ID: marinedrugs-2191317

Type of manuscript: Review. Title: Chitosan-based biomaterials: insights into chemistry, properties, devices, and their biomedical applications

Recommendation: minor revision.

This is an interesting review article in which the authors present an overview of chitosan properties and its chemical functionalization, focusing on chitosan-based biomaterials such as hydrogels, organic-inorganic hybrids, Layer-by-Layer assemblies, and bio-inks.

Overall, the manuscript is well-written, meets the criteria of the Journal and the Special Issue and can be published after minor revision.

Line 24.  biomedical arena – biomedical field or sector sounds better.

Line 207. several chemistries – sounds awkward. Perhaps chemical mechanisms or strategies fit better.

Line 425. Chitosan being a linear polymer, cross-linking is usually needed.. - the sentence should be rephrased.

Line 427. to achieve the required “performances” – to achieve the required “functionalities”

Line 432. an appropriate “choice” of cross-linking chemistry – appropriate “design” or “mechanism”

Figure 1. Check the right bracket in the chemical structure of chitin. Perhaps should be moved slightly to the right.

Figure 3 is blurry.

Scheme 7. “simultaneous approach. .” the two dots in the caption.

Scheme 8, Scheme 9, Scheme 10. Blurry cross-linking symbol.

Author Response

AUTHOR’S REPLY.
We thank the reviewer for his/her comments that helped to improve the quality of the manuscript. 
In the following, a point by point reply to his/her suggestions

Line 24.  biomedical arena – biomedical field or sector sounds better.
Amended in text as suggested 

Line 207. several chemistries – sounds awkward. Perhaps chemical mechanisms or strategies fit better.
Amended in text as suggested

Line 425. Chitosan being a linear polymer, cross-linking is usually needed.. - the sentence should be rephrased.
Amended in text as suggested

Line 427. to achieve the required “performances” – to achieve the required “functionalities”
Amended in text as suggested

Line 432. an appropriate “choice” of cross-linking chemistry – appropriate “design” or “mechanism”
Amended in text as suggested

Figure 1. Check the right bracket in the chemical structure of chitin. Perhaps should be moved slightly to the right.
Fixed as suggested

Figure 3 is blurry.
Fixed as suggested

Scheme 7. “simultaneous approach. .” the two dots in the caption.
Amended in text as suggested

Scheme 8, Scheme 9, Scheme 10. Blurry cross-linking symbol.
Fixed as suggested

Reviewer 3 Report

Review report manuscript ID  2191317

Title: Chitosan-based biomaterials: insights into chemistry, properties, devices, and their biomedical applications

This review aims at giving an overview on chitosan properties, chemical functionalization, and the innovative biomaterials obtained  thereof with  focus on the bottom-up strategies to process a wide array of chitosan-based biomaterials, given the wide literature that appeared in the last years.

Comments:

The authors made huge effort to compile this review based on many publications and patents published from 1990 to 2020. As there are numerous papers dealing with this subject, the authors are asked to highlight the novelty of their approach. Moreover,  I believe that the article should be focus on systematic review, maybe using PRISMA diagram, as there plenty of narrative reviews in this field.

In terms of chitosan film coatings used to increase shelf-life of fresh products (r370-375),  I recommend to refer at very recent paper titled: Effects of Different Edible Coatings on the Shelf Life of Fresh Black Mulberry Fruits (Morus nigra L.) DOI10.3390/agriculture12071068. Also, in order to highlight the importance and advantages of  chitosan-based materials as nanocarriers, please refer to a very recent work https://doi.org/10.3390/polym14173545.

I recommend the publication after minor adjustments.

Author Response

Please find a point by point reply to reviewer's comments.

REVIEWER COMMENT: The authors made huge effort to compile this review based on many publications and patents published from 1990 to 2020. As there are numerous papers dealing with this subject, the authors are asked to highlight the novelty of their approach. 

AUTHOR’S REPLY. We thank the reviewer for this comment. The novelty of the approach of the review has been highlighted adding an additional sentence in the abstract and in the introduction, lines 168-175: “Differently from other reviews about chitosan, this review will be organized in sections defined as a function of material typology, rather than as a function of the targeted bi-omedical application. Strategies towards CHT-based hydrogels, organic-inorganic hybrids, Layer- by-Layer systems and (bio)inks will be considered; organic-inorganic hybrids, Layer-by-Layer and inks are commonly overlooked in reviews dealing with chitosan. Particular emphasis will be given to the chemistry beyond chitosan material design, highlighting limitations and opportunities.”

REVIEWER COMMENT: Moreover,  I believe that the article should be focus on systematic review, maybe using PRISMA diagram, as there plenty of narrative reviews in this field.

AUTHOR’S REPLY. We appreciate the reviewer’s comment and we agree with his/her point of view about the richness of narrative reviews about chitosan and lack of systematic reviews (including PRISMA diagram).  We’ll keep this suggestion in mind for future reviews, however we think that the present work, organized as a narrative review for submission, requires major and in depth revisions in order to be converted into a high quality systematic review, that we feel fall out of the minor revisions requested by all reviewers and by the editorial office.  We hope that the previous reply to the reviewer may help in appreciating the value of the revised narrative review we’re proposing. 

REVIEWER COMMENT: In terms of chitosan film coatings used to increase shelf-life of fresh products (r370-375),  I recommend to refer at very recent paper titled: Effects of Different Edible Coatings on the Shelf Life of Fresh Black Mulberry Fruits (Morus nigra L.) DOI10.3390/agriculture12071068. Also, in order to highlight the importance and advantages of  chitosan-based materials as nanocarriers, please refer to a very recent work https://doi.org/10.3390/polym14173545.

AUTHOR’S REPLY. We appreciate reviewers' suggestions. In order to highlight the importance and advantages of  chitosan-based materials as nanocarriers, the very recent work https://doi.org/10.3390/polym14173545 was added as suggested (line 475) 

The additional paper, despite being extremely interesting in the field of edible coatings, is an original article not dealing with chitosan, despite some recent papers about chitosan-based coatings have been cited by the authors. Since the literature included in the present review is already particularly extended, we think that the proposed reference does not add value to the review bibliography.

Reviewer 4 Report

The topic submitted on chitosan has already large reviews published earlier but does cover the latest significant research data to the existing field of research. The article is not very well articulated and needs English language revisions and even formatting of the manuscript as per the MDPI guidelines. The manuscript needs to be checked for proper numbering and citation of references. The introduction needs to be enlarged with more background. The abstract and conclusion should include a sentence proposing the future direction of the presented research topic of at least 1-2 lines. Also the commercial aspects of how patients and pharmaceutical industries can benefit from chitosan based products needs to be added.  Most of the sections discussed lack citing the references from the latest and previous studies. Below are a few suggestions the authors are requested to discuss and cite in appropriate sections.

International journal of biological macromolecules 109 (2018): 273-286.

 International Journal of Biological Macromolecules165, 1924-1938.

Environmental Chemistry Letters 18, no. 3 (2020): 577-594.

 European Polymer Journal 147 (2021): 110326.

Journal of drug targeting 27, no. 4 (2019): 379-393.

Journal of Drug Delivery Science and Technology 64 (2021): 102579

Gels7(4), p.253.

Nanoscale 5, no. 8 (2013): 3103-3111.

 Polysaccharides2(2), pp.519-537.

Author Response

Please find a point by point reply to reviewer's comments.

REVIEWER COMMENT: The topic submitted on chitosan has already large reviews published earlier but does cover the latest significant research data to the existing field of research. The article is not very well articulated and needs English language revisions and even formatting of the manuscript as per the MDPI guidelines.  The manuscript needs to be checked for proper numbering and citation of references. 

AUTHOR’S REPLY. We appreciate reviewer comments about formatting, which is a relevant issue in defining the final quality of the manuscript. For manuscript and bibliography preparation we used the MDPI template and the suggested Zotero citation manager software, setting the Marine Drugs style. English language has been checked throughout the manuscript.

REVIEWER COMMENT: The introduction needs to be enlarged with more background. Also the commercial aspects of how patients and pharmaceutical industries can benefit from chitosan based products needs to be added. 

AUTHOR’S REPLY. We appreciate reviewer suggestion; the introduction has been enlarged, by adding the benefits of patients and of the pharmaceutical sector (lines 143-151): “Patients and pharmaceutical industries will benefit from the chitosan-based products that will enter into the market, due to their potentialities as drug delivery systems, pharmaceutical formulation components, and as active ingredients for different medi-cal treatments.  Formulations and drug delivery systems containing chitosan for ex-ample can help in the delivery of therapeutically relevant active proteins (i.e. growth factors), and peptides, nucleic acids and genes. The patients would benefit in the sec-tors of regenerative medicine, oncology, dermatology, ophthalmology, dentistry and many others both from a therapeutic and diagnostic point of view.”

REVIEWER COMMENT: The abstract and conclusion should include a sentence proposing the future direction of the presented research topic of at least 1-2 lines. 

AUTHOR’S REPLY.  We appreciate reviewer suggestion; the abstract now includes a couple of lines inspiring the reader towards future directions (lines 25-27: “...will be covered aiming to elucidate and inspire the community to keep on exploring the unique features and properties imparted by chitosan to develop advanced biomedical devices”), and the conclusions include a sentence highlighting future directions to improve scientific knowledge and applications of chitosan (lines 1537-1545): “The environmental and production costs will be key issues for chitosan to enter the market as competitive substitute of fossil-based feedstocks for the production of innovative materials for several applications. In recent years, studies are emerging aimed at the detailed life cycle assessment (LCA) of chitosan production, specifically focusing on environmental and economic viability [9]. We are sure that in the next few years CHT  will strongly contribute to unprecedented advancements in the material sciences and biomedical fields, thanks to the progresses in alleviating the key issues that are still a limitation to chitosan exploitation.”

REVIEWER COMMENT: Most of the sections discussed lack citing the references from the latest and previous studies. Below are a few suggestions the authors are requested to discuss and cite in appropriate sections.

AUTHOR’S REPLY.  We appreciate reviewer suggestions towards the improvement of the bibliographic section.

International journal of biological macromolecules 109 (2018): 273-286.  Added as suggested  (line 475)

International Journal of Biological Macromolecules, 165, 1924-1938.  Added as suggested  (line 142)

Environmental Chemistry Letters 18, no. 3 (2020): 577-594. Added as suggested  (line 475)

European Polymer Journal 147 (2021): 110326. Added as suggested  (line 475)

Journal of drug targeting 27, no. 4 (2019): 379-393. Added as suggested  (line 475)

Journal of Drug Delivery Science and Technology 64 (2021): 102579  Added as suggested  (line 1380)

Gels, 7(4), p.253. Added as suggested  (line 475)

Nanoscale 5, no. 8 (2013): 3103-3111. Added as suggested  (line 475)

Polysaccharides, 2(2), pp.519-537. Added as suggested  (line 475)